# Katanin-Dependent Microtubule Ordering in Association with ABA Is Important for Root Hydrotropism

**DOI:** 10.3390/ijms23073846

**Published:** 2022-03-31

**Authors:** Rui Miao, Wei Siao, Na Zhang, Zuliang Lei, Deshu Lin, Rishikesh P. Bhalerao, Congming Lu, Weifeng Xu

**Affiliations:** 1Joint International Research Laboratory of Water and Nutrient in Crops and College of Resource and Environment, Center for Plant Water-Use and Nutrition Regulation and College of Life Sciences, Fujian Agriculture and Forestry University, Jinshan, Fuzhou 350002, China; miaorui2011@126.com (R.M.); setwind0820@gmail.com (W.S.); 15737315136@163.com (N.Z.); 1200525014@fafu.edu.cn (Z.L.); 2Basic Forestry and Proteomics Research Center, Fujian Provincial Key Laboratory of Haixia Applied Plant Systems Biology, College of Life Science, Fujian Agriculture and Forestry University, Jinshan, Fuzhou 350002, China; deshu.lin@fafu.edu.cn; 3Beijing Advanced Innovation Center for Tree Breeding by Molecular Design, Beijing Forestry University, Beijing 100083, China; rishi.bhalerao@slu.se; 4Umea Plant Science Centre, Department of Forest Genetics and Plant Physiology, Swedish University of Agricultural Sciences, 901 87 Umea, Sweden; 5State Key Laboratory of Crop Biology, College of Life Sciences, Shandong Agricultural University, Tai’an 271018, China; cmlu@sdau.edu.cn

**Keywords:** cortical microtubule arrays, oryzalin, KATANIN, abscisic acid, root hydrotropism

## Abstract

Root hydrotropism refers to root directional growth toward soil moisture. Cortical microtubule arrays are essential for determining the growth axis of the elongating cells in plants. However, the role of microtubule reorganization in root hydrotropism remains elusive. Here, we demonstrate that the well-ordered microtubule arrays and the microtubule-severing protein KATANIN (KTN) play important roles in regulating root hydrotropism in Arabidopsis. We found that the root hydrotropic bending of the *ktn1* mutant was severely attenuated but not root gravitropism. After hydrostimulation, cortical microtubule arrays in cells of the elongation zone of wild-type (WT) Col-0 roots were reoriented from transverse into an oblique array along the axis of cell elongation, whereas the microtubule arrays in the *ktn1* mutant remained in disorder. Moreover, we revealed that abscisic acid (ABA) signaling enhanced the root hydrotropism of WT and partially rescued the oryzalin (a microtubule destabilizer) alterative root hydrotropism of WT but not *ktn1* mutants. These results suggest that katanin-dependent microtubule ordering is required for root hydrotropism, which might work downstream of ABA signaling pathways for plant roots to search for water.

## 1. Introduction

Hydrotropism is a root tropic response in which roots grow toward an area with higher water potentials [1]. Recent articles reported that phytohormones such as abscisic acid (ABA), brassinosteroids (BRs), and cytokinins (CKs) participate in regulating root hydrotropism in Arabidopsis [2,3,4,5,6,7,8]. The stress hormone ABA plays an important role in response to water stress, and the ABA signalosome (PYR/PYL/RCAR-PP2Cs-SnRK2s) is involved in root hydrotropic responses in Arabidopsis [2,3,6,9,10]. The ABA receptor sextuple mutant *pyr1pyl1pyl2pyl4pyl5pyl8* (*112458*) exhibits a reduced root hydrotropic response [2], whereas the clade A protein phosphatase type 2C (PP2Cs) quadruple mutant *hab1-1abi1-2abi2-2pp2ca-1* (*Qabi2-2*) was found to exhibit enhanced root hydrotropism caused by the relief of the ABA INSENSITIVE 1 (ABI1)-mediated inhibition of plasma membrane H^+^-ATPase 2 (AHA2) [2,6]. Based on the acid-growth theory, apoplastic H^+^ efflux might loosen cell walls to assist hydrotropic responses in Arabidopsis roots. In accordance with this hypothesis, the *snrk2.2 snrk2.3* double mutant, which lacks the key positive regulators of ABA signaling, was insensitive to hydrotrostimulation in Arabidopsis roots [3]. Thus far, autophagy, reactive oxygen species (ROS) signaling, and calcium signaling were also found to be involved in root hydrotropic responses [11,12,13].

The regulatory mechanism of the dynamic instability of cortical microtubules, switching between shrinkage and growth by polymerization, is conserved in eukaryotes [14,15]. In higher plants, the rearrangements of cortical microtubule arrays occur during cell growth and division, and the cortical microtubule arrays direct the movement of the plasma membrane-localized cellulose synthase for cell-wall patterning [16,17,18,19,20,21]. Previous studies have demonstrated that cortical microtubule dynamics are important for the directional growth of roots upon environmental stimuli, such as light, temperature, and gravity [22,23,24]. After gravistimulation, auxin is rapidly redistributed to the lower side of the root, leading to the reorientation of microtubule arrays toward a longitudinal direction of the root, further resulting in growth inhibition in the lower side and, thus, root bending [23,25]. Endogenous or exogenous application of auxin leads to a rapid rearrangement of cortical microtubules toward a longitudinal direction in etiolated hypocotyls and roots, resulting in growth inhibition in higher plants [26]. In gravistimulated *Physcomitrella patens*, a microtubule reoriented and gathered mainly on the lower flank of the tips of caulonemal filaments [27]. Taken together, the rearrangement of microtubule arrays is proposed as a way to direct growth toward the gravity vector via the regulation of auxin gradients in plant roots [22,26]. However, the role of cortical microtubules in controlling root hydrotropic responses remain an issue to be elucidated.

Microtubule-associated proteins (MAPs) bind directly to microtubules and regulate microtubule organization and the direction of cell expansion in plants [28]. MAPs control and fine tune microtubule dynamic instability, which is essential for the orientation of cell and directional growth [29]. The well-characterized microtubule-severing protein KATANIN, an AAA ATPase protein, disassembles microtubule minus-ends from the nucleation sites to depolymerize the minus-ends of cytoskeletal polymers and mediates microtubule spatial organization [28,29]. *KATANIN1* (*KTN1*) maintains cortical microtubule ordering in Arabidopsis, and the loss-of-function mutant *ktn1-2* shows disordered cortical microtubule arrays and reduced cell division [30]. *KTN1* is also critical for cells in response to mechanical stress, which amplifies the differences in the rate of cell elongation between neighboring cells in Arabidopsis [31].

In this study, we show that *KTN1* plays an important role in root hydrotropism. The cortical microtubule arrays in wild-type (WT) roots were rearranged after hydrostimulation treatment, but the cortical microtubule arrays in the *ktn1* mutant were not able to be reorganized, resulting in a defect in root hydrotropic bending. In addition, ABA signaling enhanced the root hydrotropism of WT but not *ktn1* mutants, indicating that cortical microtubule ordering might work downstream of ABA signaling pathways for root hydrotropic responses.

## 2. Results and Discussion

### 2.1. Hydrotropic Bending of WT Col-0 Roots Is Reduced by Oryzalin Treatment

Root hydrotropism is the directional growth of roots bending toward soil moisture. Baral et al. [32] recently showed that tissue bending requires microtubule array rearrangements in Arabidopsis. In this study, we first examined the effects of the microtubule destabilizer oryzalin (100 nM and 200 nM), which depolymerizes microtubules and decreases the number of microtubule arrays [33], on the root hydrotropic bending of wild-type (WT) ecotype Col-0 in the hydrotropism assay [5]. During a time course of 24 h, the hydrotropic bending curvature of WT roots reached its maximum after 11 h of hydrostimulation in our assay (Figure 1A). We observed the average root-bending angles of approximately 19, 29, and 18° after 5, 11, and 24 h of hydrostimulation, respectively. Interestingly, when 100 nM of oryzalin was applied, the hydrotropic curvature of WT roots reached its maximum at 5 h and then remained the same until the end of our assays. The average root-bending angles of 16, 15, and 15° were observed after 5, 11, and 24 h of hydrostimulation, respectively. As a result, the oryzalin-treated roots showed significantly reduced hydrotropic bending at 11 h compared with the roots under mock treatment condition (Figure 1A). In addition, there were no significant differences in root growth between the 100 nM oryzalin-treated and untreated roots within 11 h, suggesting that attenuated hydrotropic bending was not caused by retarded root growth. The inhibition effect of oryzalin treatment on root growth was observed only after 11 h of oryzalin treatment, at which time no significant difference on the hydrotropic curvatures between treatments can be observed (Figure 1A). These results indicated that the inhibitory effects of oryzalin on the root hydrotropic response occurred before root growth was attenuated, suggesting that proper organization of microtubule arrays is essential for achieving the full bending of hydrostimulated roots. On the other hand, the gravitropic bending of WT roots was not affected in the presence of 100 nM oryzalin (Figure 1B), which is consistent with previous publications [34,35,36]. The untreated and oryzalin-treated roots displayed similar bending angles, which are on average approximately 50, 80, and 80° after 5, 11, and 24 h of gravistimulation. Additionally, root growth was severely attenuated upon the 200 nM oryzalin treatment, which is not a favorable condition to justify the role of oryzalin in either root hydrotropism or gravitropism (Figure 1A,B). Taken together, these results demonstrated that microtubule depolymerization caused by oryzalin treatment had greater effects on root hydrotropism than root gravitropism in Arabidopsis.

### 2.2. Katanin Is Required for Root Hydrotropism in Arabidopsis

Since katanin, a microtubule severing enzyme, is known to play an important role as a catalyzer of tissue bending, we performed hydrotropic assays by using Arabidopsis *KTN1* mutants *ktn1-4* and *ktn1-6*, which exhibit compromised microtubule turnover [30,33], to examine if katanin-dependent microtubule organization is involved in root hydrotropism. The results showed that in the 5 days after germination (dag) *ktn1-4* and *ktn1-6* seedlings displayed significantly reduced root hydrotropic responses compared with that of WT (Figure 2A), but the root gravitropic curvatures in *ktn1-4* and *ktn1-6* seedlings were not significantly influenced (Figure 2B). The root growth of *ktn1-4* and *ktn1-6* seedlings is also much slower than that of WT (Figure 2A,B). These results suggested that microtubule dynamics is critical for hydrotropic responses but not for gravitropic responses in Arabidopsis roots.

To understand how cortical microtubules are rearranged during root hydrotropic responses, we next analyzed cortical microtubule ordering in roots of WT and the *ktn1* mutant. The transgenic Arabidopsis lines expressing α-tubulin 6 fused to green fluorescent protein driven by a *35S* promoter in Col-0 (*GFP-TUA6*/Col-0) and *ktn1* background (*GFP-TUA6*/*ktn1*) were used to visualize cortical microtubules by using confocal microscopy [30,37]. After hydrostimulation treatments, the cortical microtubules in the cells of the root elongation zone were rearranged from perpendicular arrays into an oblique orientation along the root growth axis, whereas the orientation of the cortical microtubule arrays in the root maturation and transition zones remained the same (Figure 3A,C,E and Appendix A). On the contrary, cortical microtubules in the root cells of the *ktn1* mutant consistently showed disordered organization before and after hydrostimulation and were even more transverse in the hydrotropism assay (Figure 3B,D,F). Taken together, with the reduced hydrotropic responses in the *ktn1* mutant, these data suggest that cortical microtubule ordering is required for root hydrotropism.

### 2.3. Katanin-Regulated Microtubule Ordering Might Work Downstream of ABA in Root Hydrotropism

To investigate the correlation between ABA signaling pathways and cortical microtubule dynamics during root hydrotropic response, we analyzed the root hydrotropic response of WT, *Qabi2-2* (*hab1-1abi1-2abi2-2pp2ca-1* quadruple mutant), *ktn1-4*, and *miz1* mutants in the presence of 1 μM ABA and 100 nM oryzalin upon the hydrotropism assay. *MIZU-KUSSEI1* (*MIZ1*) was essential for root hydrotropism [1], and the *Qabi2-2* quadruple mutant showed increased root hydrotropic responses [2,6]. We observed that the root curvature of the WT, but not the *Qabi2-2* mutant, was enhanced by 1 μM ABA treatment. This may be because *Qabi2-2* has already stayed in an ABA active status, and extra ABA (1 μM) application cannot promote root curvature further in the hydrotropism assay (Figure 4D). Nevertheless, root hydrotropic bending in WT and the *Qabi2-2* mutant was significantly reduced by the application of oryzalin (Figure 4D). Interestingly, the supplements with indicated concentrations of ABA or oryzalin had no effect on root gravitropic bending in WT, *Qabi2-2*, *ktn1-4*, and *miz1* mutants (Figure 4E). These results indicated that cortical microtubule ordering mainly functions in ABA-enhanced root hydrotropism but not in root gravitropism.

To confirm the roles of cortical microtubule arrays in regulating ABA signaling pathways during root hydrotropic responses, we performed high-throughput RNA sequencing by a BGISEQ platform by using Arabidopsis root tips (<5 mm from the root cap) of Col-0 WT and *ktn1-4* mutant seedlings in the absence or presence of 1 μM ABA as materials, which produced more than 1.08 G clean reads mapped onto the Arabidopsis genome sequence, and at least 25,868 gene expressions were detected from root tips of the WT and *ktn1-4* mutant. A total of 44 and 700 differentially expressed genes (DEGs) (probability ≥ 0.95) were found in roots of WT and the *ktn1-4* mutant on the split agar-based hydrotropism assay, respectively (Figure 4A–C). Among these genes, 24 and 429 of DEGs displayed higher expressions, whereas 20 and 271 genes demonstrated lower expression in roots of WT and the *ktn1-4* mutant under the hydrotropism assay, respectively (Figure 4A,B). A Venn diagram analysis of these DEGs exhibited 13 common DEGs between Col-0-NM (NM: normal medium) vs. Col-0-HT (HT: hydrostimulation treatment) and *ktn1*-NM vs. *ktn1*-HT (Figure 4C). In the presence of 1 μM ABA, the ABA-related DEGs between WT Col-0 and *ktn1-4* root tips were observed after hydrostimulation treatment (Figure 4F). Among them, interestingly, the application of 1 μM ABA reduced the expression levels of the negative regulators (i.e., *CLO4* (AT1G70670)) but enhanced the expression levels of the positive regulators (i.e., *FBA2* (AT4G38970)) in the ABA signaling pathway in WT, compared with those of the *ktn1-4* mutant in the hydrotropism assay (Figure 4F and Figure 5A,B), indicating that ABA treatment stimulated root hydrotropism through the ABA signal transduction pathway in WT but not the *ktn1-4* mutant. On the one hand, the mRNA level of *CLO4* (*Caleosin 4*) is downregulated in response to ABA and salt stress [38]. The *clo4* mutants were more tolerant to drought than the wild type, whereas *CLO4*-overexpressing (Ox) lines showed hyposensitivity to those abiotic stresses [38]. On the other hand, the mRNA level of *FBA2* (*fructose 1,6-biphosphate aldolase 2*) is upregulated following exposure to exogenous ABA, high salinity, and drought and may play key roles in ABA and stress signaling in plants [39]. Moreover, the Tyr phosphorylation level of FBA is modulated by ABA [40]. These results suggest that the lack of katanin might inhibit the gene expression of ABA-related signaling. Taken together, these data suggested that katanin-regulated microtubule ordering is essential for root hydrotropism and might work downstream of the ABA signaling pathway during root hydrotropic bending.

## 3. Materials and Methods

### 3.1. Plant Material and Growth Conditions

Seeds of *Arabidopsis thaliana* wild-type (WT) ecotype Columbia-0 (Col-0) and *ktn1* mutants (*ktn1-4* and *ktn1-6*; AT1G80350), which were acquired from Dr. Deshu Lin and Dr. Rishikesh P. Bhalerao, respectively, were surface-sterilized with ~5% (*v*/*v*) sodium hypochlorite for 3 min. Subsequently, seeds were washed five times with sterile water and sown on plates containing 1/2 Murashige and Skoog (MS) medium with 0.8% (*w*/*v*) and agarose 1% (*w*/*v*) sucrose. *ktn1-4* and *ktn1-6* mutants are in the Col-0 background. Plates were kept at 4 °C for 2 days for vernalization. After that, the seeds were grown vertically for 5 days and then transferred to 1/2 MS medium in the absence or presence of 100 nM oryzalin or 1 μM ABA for vertical growth lasting 11 or 24 h in the hydrotropism assay, as described by Miao et al. [5]. The Arabidopsis seedlings were placed in growth chambers at light dark cycles (16 h light (23 °C) and 8 h dark (21 °C)).

### 3.2. RNA-Sequencing Assay

For RNAsequencing, WT and *ktn1-4* seedlings were germinated in 1/2 MS medium for 5 days and then transferred to the hydrotropism assay in the absence or presence of 1 μM ABA for vertical growth lasting for 11 h. Approximately 1.0 g of root tips (<5 mm from the root cap) was harvested, and each accession included three biological replicates. The collected samples were immediately frozen in liquid nitrogen. Then, total RNA was extracted from root tissues by a TRIzol reagent (Invitrogen, Carlsbad, CA, USA). The quality and quantity of the total RNA were determined by an Agilent 2100 Bioanalyzer (Agilent Technologies, Santa Clara, CA, USA). After that, first-strand cDNA was synthesized and sequenced according to Miao et al. [5]. Six libraries, Col-0-NM [seedlings grown in the normal medium (NM)], Col-0-HT (hydrostimulation treatment), *ktn1*-NM, and *ktn1*-HT were constructed. The RNA-sequencing data generated in this study have been deposited at NCBI in the Sequence Read Archive (SRA) database under the accession number PRJNA610935 (https://www.ncbi.nlm.nih.gov/sra/PRJNA610935; accessed on 18 January 2022).

### 3.3. Quantitative (Q) RT-PCR

The RNA was isolated for 5 days after germination (dag) with WT and *ktn1* seedlings after transferring them to NM or the hydrotropism assay for vertical growth lasting 11 h by using the TRIzol reagent (Product No. 93289, Sigma-Aldrich, St. Louis, MO, USA). The first-strand cDNA was generated by 1 μg RNA, the oligo(dT) 15 primer, and M-MLV reverse transcriptase (Promega, Madison, MI, USA). Each reaction contains the same aliquot of the first-strand cDNA as a template with gene-specific primers. *Actin2* was used as an internal control and analyzed by the same cDNA and SYBR green, which includes four technical replicates. The primers are listed in Appendix A.

### 3.4. Drug Treatments

Stock solutions (20 mM) of the microtubule-disrupting drug oryzalin were dissolved in dimethyl sulfoxide (DMSO) and added to MS medium at the final concentrations indicated in the text. The *A. thaliana* ecotype Col-0 and *ktn1* loss-of-function mutant were sown in 1/2 MS medium (1% (*w*/*v*) sucrose and 0.8% (*w*/*v*) agarose) in vertically positioned Petri dishes containing the microtubule-disrupting drug oryzalin and grown for the indicated times before the following experiments. DMSO treatment was used as a control.

### 3.5. Microscopy

The transgenic *A. thaliana* lines expressing *GFP-TUA6* were used to visualize microtubules in root cortical cells. The *ktn1* mutant was crossed with *GFP-TUA6* (Lin et al. [30]). In the split/agar-based hydrotropism assays, the cortical microtubules in roots of both genotypes were fixed for 1 h at the indicated time in a microtubule-stabilizing buffer including 8% (*v*/*v*) formaldehyde according to Hou et al. [35] with a minor revision and observed by using a Leica SP8 confocal laser-scanning microscope (Leica, Heidelberg, Germany) with an excitation at 488 nm and emission at 535 nm.

## Figures and Tables

**Figure 1 ijms-23-03846-f001:**
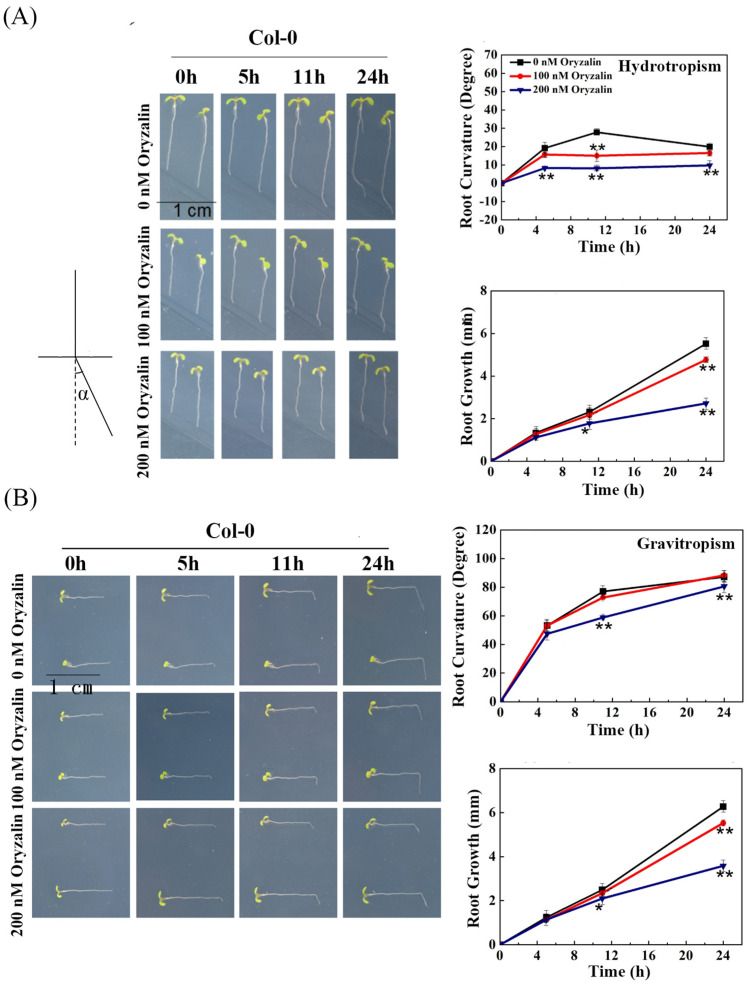
Oryzalin treatments decreased the root hydrotropic response of wild–type (WT) ecotype Col-0. (**A**) Root hydrotropic response and growth of ecotype WT in the absence or presence of 100 nM or 200 nM oryzalin (microtubule destabilizer: microtubule–disrupting chemical). Schematic representation of the root curvature angle (α) in the hydrotropism assay was indicated. (**B**) Root gravitropic response and growth of ecotype WT roots in the absence or presence of 100 nM or 200 nM oryzalin. Error bars indicate the SE; *n* ≥ 36. Asterisk indicates significant difference from the WT at * *p* < 0.05 or ** *p* < 0.01 (Student’s *t*-test).

**Figure 2 ijms-23-03846-f002:**
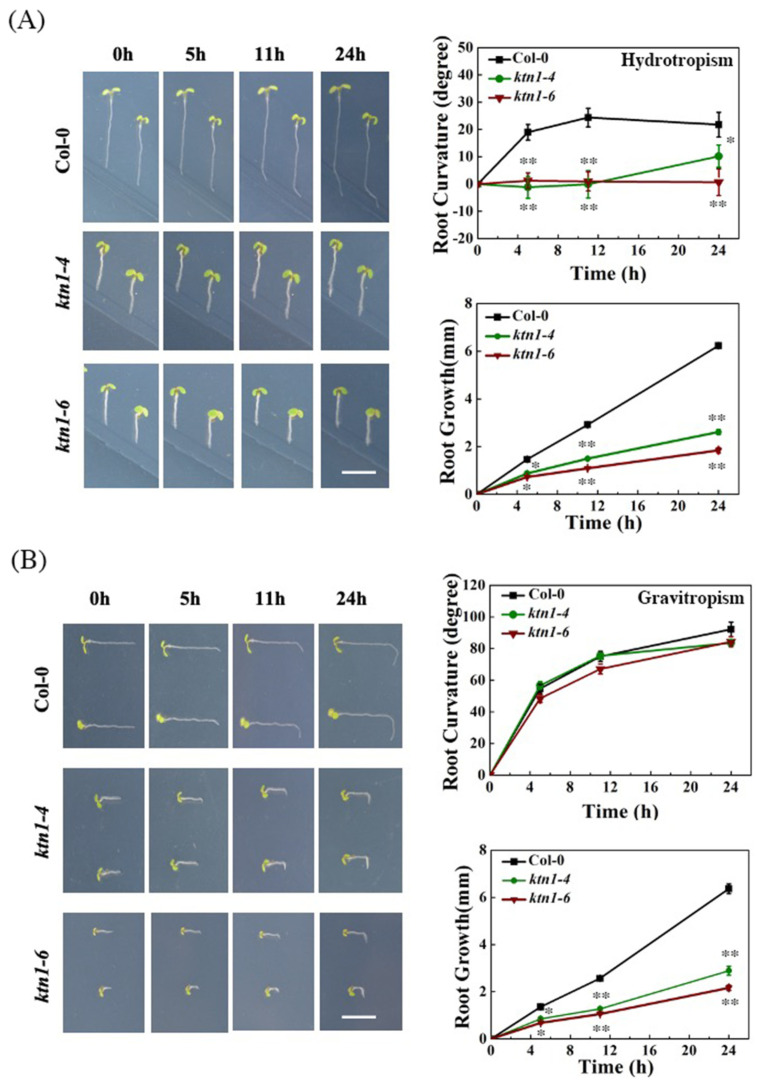
Arabidopsis *KATANIN* mutants (*ktn1*) show reduced root hydrotropism. (**A**) Measurements of root curvatures and growth rates of *ktn1-4* and *ktn1-6* mutants after hydrostimulation treatment for 0, 11, and 25 h. (**B**) Measurements of root curvatures and growth rates of *ktn1-4* and *ktn1-6* mutant after gravistimulation treatment for 0, 11, and 25 h. Scale bar = 1 cm. Error bars indicate the SE; *n* ≥ 36. Asterisks show significant difference from the wild type at * *p* < 0.05 or ** *p* < 0.01 (Student’s *t*-test).

**Figure 3 ijms-23-03846-f003:**
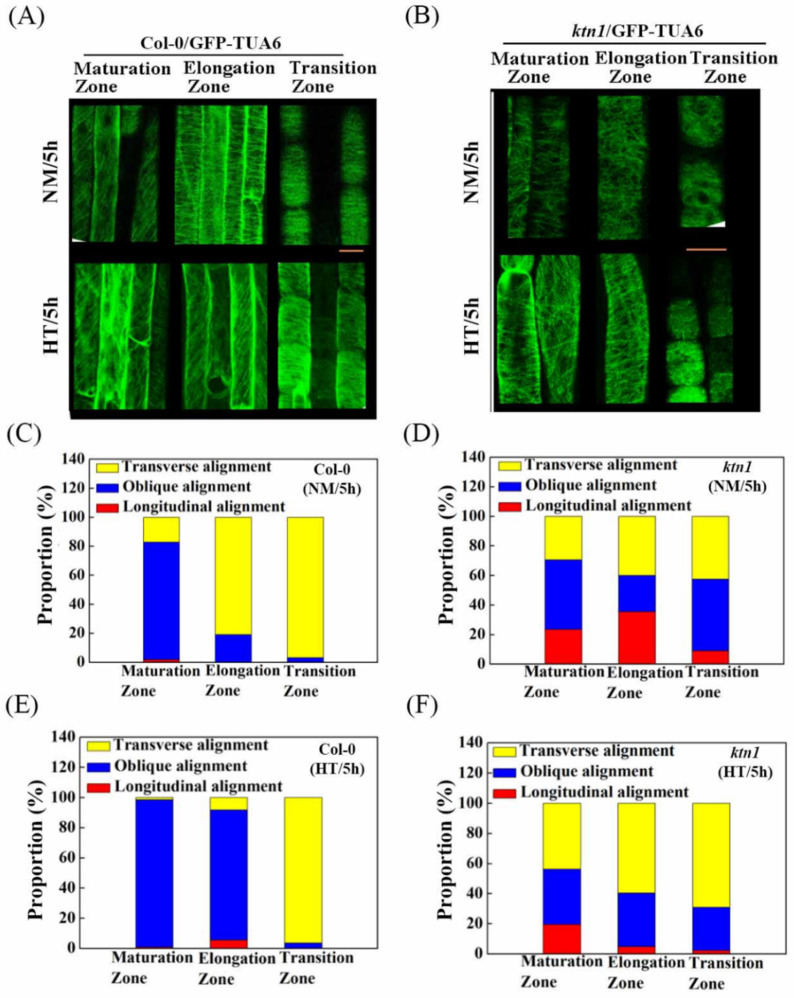
Cortical microtubule arrays in the root cells of WT Col-0 but not the *ktn1* mutant are reoriented after hydrostimulation treatment. Cortical microtubules were observed in cortical cells of root maturation (left), elongation (middle), and transition (right) zones from WT (**A**) and the *ktn1* mutant (**B**) under control and hydrostimulation treatment (HT). Quantification of root cortical cell microtubules’ orientation in WT under control (**C**) and hydrostimulation treatment (HT) (**E**) for 5 h (*n* > 30 cells of each line). Quantification of root cortical cell microtubule orientation in the *ktn1* mutant under control (**D**) and hydrostimulation treatment (HT) (**F**) for 5 h (*n* ≥ 30 cells of each line). Scale bar = 10 μm. Cortical microtubule orientation was measured by using ImageJ v.1.46 (National Institutes of Health, Bethesda, MD, USA).

**Figure 4 ijms-23-03846-f004:**
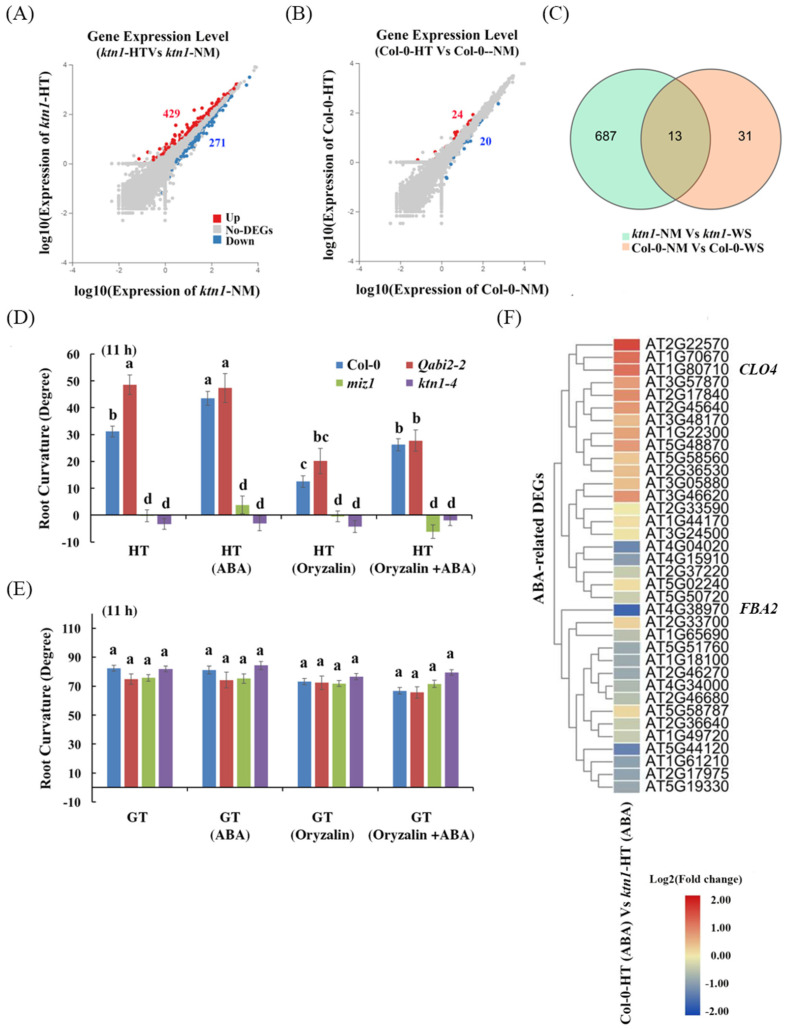
ABA enhances root curvatures of WT Col-0 and partially rescues the oryzalin-alterative root curvatures of WT but not *miz1* and *ktn1-4* mutants after hydrostimulation treatment. (**A** and **B**) Scatter plots of transcript abundance of *ktn1* (**A**) and Col-0 (**B**) roots in the hydrotropism assay for two days. Red, blue, and gray dots represent upregulated, downregulated, and not differently expressed genes, respectively (probability ≥ 0.95 and log_2_Ratio ≥ 1). (**C**) Venn diagram showing the number of DEGs of *ktn1* and Col-0 in the hydrotropism assay. (**D**) Measurements of root curvatures of ABA and oryzalin (a microtubule destabilizer) treatments on WT, *Qabi2-2* (an ABA sensitive mutant), *miz1* (a lost hydrotropism mutant), and *ktn1-4* (a microtubule-associated protein (MAP) mutant) after hydrostimulation treatment (HT). (**E**) Measurements of root curvatures of ABA and oryzalin treatments on WT, *Qabi2-2*, *miz1*, and *ktn1-4* mutants after gravistimulated treatment (GT). Error bars indicate SE; *n* = 36. Lower case letters show significant difference at *p* < 0.05 (Tukey’s test). (**F**) Heat map showing ABA-related differentially expressed genes (DEGs) in Col-0-HT (ABA) vs. *ktn1*-HT (ABA). Red = high transcript level; blue = low transcript level. The transcriptional profiles of ABA-related gene expression values (log2Ratio values) were analyzed by using the heat map command of the R language, which are displayed as colors ranging from red to blue as shown in the key.

**Figure 5 ijms-23-03846-f005:**
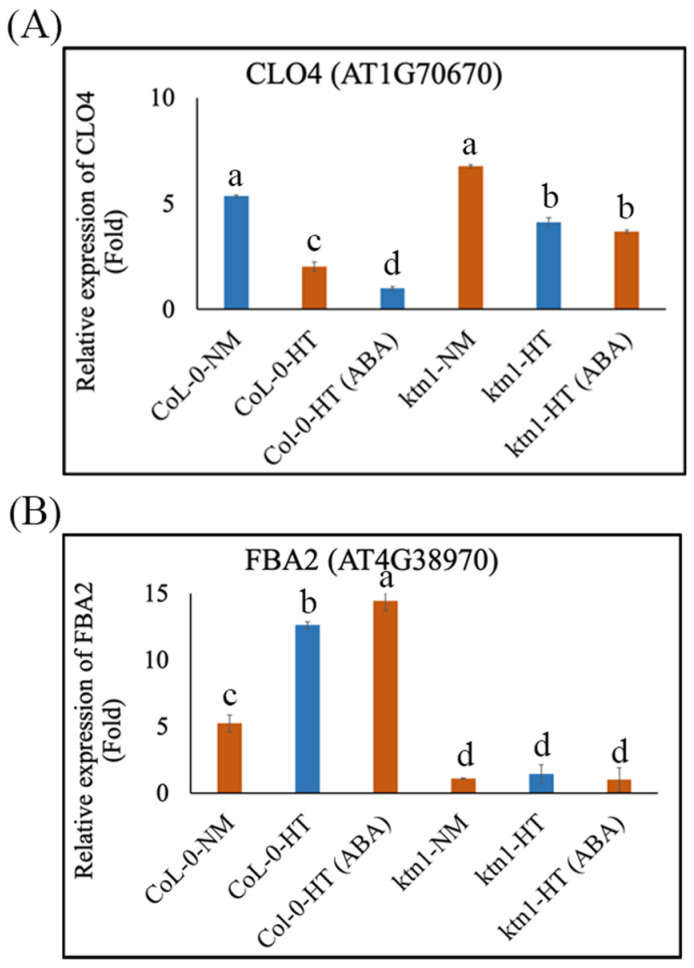
Relative gene expression values of ABA-related genes in roots of Col-0 and the *ktn1* mutant upon the hydrotropism assay (**A**) Relative gene expression values of *CLO4* (AT1G70670, a negative regulator in ABA signaling) in Col-0-NM, Col-0-HT, Col-0-HT (ABA), *ktn1*-NM, *ktn1*-HT, and *ktn1*-HT (ABA). (**B**) Relative gene expression values of *FBA2* (AT4G38970, a positive regulator in ABA signaling) in Col-0-NM, Col-0-HT, Col-0-HT (ABA), *ktn1*-NM, *ktn1*-HT, and *ktn1*-HT (ABA). The values are means, and error bars show ± SE of four technical replicates from three independent experiments (*n* = 12). Values with the same letters were not significantly different from one another (one-way ANOVA, *p* < 0.05).

## Data Availability

The datasets used or analyzed during the current study are available from the corresponding author upon reasonable request.

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
