# Peer review of "Katanin-Dependent Microtubule Ordering in Association with ABA Is Important for Root Hydrotropism"

_ijms, 2022, doi:10.3390/ijms23073846_

Round 1
Reviewer 1 Report
Root hydrotropism is a very important phenomenon. Combining this phenomenon with the Cortical microtubule orientation is crucial in understanding the role of simple mechanics in the water searching by plants. However, in this paper author demonstrate only that the well-ordered microtubule arrays play important roles in regulating root hydrotropism in Arabidopsis, the result seems to be promising and certainly, any correlation will be found in the future which will be quantified, especially the mechanistic ones.
I do not feel competent enough to evaluate the paper from the genetic point of view, but these results seem to be reasonable. However, some lacks in calculation emerge, so the paper should be improved in this area before publication. Below please find my remarks and comments.
Line 152: I do see differences between treated and untreated plants, if they are significant should be proven. It is only mentioned that the t-student test has been done, but there is a lack of curvature definition, also I cannot see the error bars. Also, I could not find the number of considered plants taken into statistical analysis. These should be clearly presented to avoid any misunderstandings.
Line 156: I do see differences between cases. Untreated is changing its curvature while the treated ones are not. Quantified data should be presented and analyzed if significant differences are present, else the conclusion in lines 157-159 is not confirmed.
Lines 161 – 164: Lack of quantified results and analysis
Line 183: Another reason can be that destabilization (oryzalin origin) is not strong enough, the MT rearrangement should emerge (ktn mutants).
It is also a bit frustrating that authors firstly refer to supplement figures instead of beginning from the first in the main text, this should be fixed.
Also, the result with 200nM oryzalin treatment was not enough considered.
Concluding, the manuscript is very interesting and promissive. If the calculation is improved and if the experimental data are clearly presented I recommend it for the publishing process.
Author Response
Line 152: I do see differences between treated and untreated plants, if they are significant should be proven. It is only mentioned that the t-student test has been done, but there is a lack of curvature definition, also I cannot see the error bars. Also, I could not find the number of considered plants taken into statistical analysis. These should be clearly presented to avoid any misunderstandings.
Response: Great thanks for your comments! Indeed, there were no significant differences in root growth between the 100 nM oryzalin treated and untreated roots within that 11 hours. The curvature definition has been demonstrated in Figure 1A at the updated manuscript. The number of considered plants taken into statistical analysis had been labelled in each figure legend i.e. “Error bars indicate the SE; n≥36” in Figure 1 legend. There are the error bars, and probably the error bars were too small to see previously.
Line 156: I do see differences between cases. Untreated is changing its curvature while the treated ones are not. Quantified data should be presented and analyzed if significant differences are present, else the conclusion in lines 157-159 is not confirmed.
Response: The quantified data have been presented and analyzed between lines 148 and 153 at the revised manuscript.
Lines 161 – 164: Lack of quantified results and analysis
Response: The quantified data have been presented and analyzed between lines 165 and 167 at the revised manuscript.
Line 183: Another reason can be that destabilization (oryzalin origin) is not strong enough, the MT rearrangement should emerge (ktn mutants).
It is also a bit frustrating that authors firstly refer to supplement figures instead of beginning from the first in the main text, this should be fixed.
Response: We have put the Figure S1 as the Figure 1 at the revised manuscript.
Also, the result with 200nM oryzalin treatment was not enough considered.
Response: the root growth was severely attenuated upon the 200 nM oryzalin treatment. Hence, it is not a good condition to justify the role of oryzalin in root hydrotropism, and we have described it between lines 168 and 169.
Reviewer 2 Report
The manuscript presents interesting findings regarding root hydrotropism and offers new insights about the possible involvement of ABA in regulation of root bending in response to moisture gradient by adjusting MTs rearrangement.
Comments –
- ktn1 mutants are shown to display significant reduction of hydrotropic root bending upon hydrostimulation, rather normal gravitropic growth but overall have significant reduced primary root growth. Lines 180-182 – “The root growth of ktn1-4 and ktn1-6 seedlings is also much slower than that of WT (Fig. 1A and B). These results suggested that microtubule dynamics is critical for hydrotropic responses but not for gravitropic responses in Arabidopsis roots”.
The reduced root growth upon control conditions suggests that the ktn1 mutation have more global effect on root growth that logically also affects hydrotropism. The normal gravitropism might indicate that at the time frame of few hours the attenuated root growth not necessarily affects the early root tropic response. However, it is still not clear whether the reduced hydrotropic response is due to the observed reduced root growth of ktn1 under control conditions, and is not directly related to hydrotropism.
- Treatment with oryzalin (0, 100 nM) of WT seedlings showed no significant effect over root hydrotropism during the initial 4 hrs of stimulation. This was shown by treating the seedlings while exposing the roots to the chemical and moisture gradient simultaneously. It woud be more informative if the seedlings were pre-treated with oryzalin prior to hydrostimulation such that the effect on MTs arrangement will be from hydrostimulation start and might provide more insights regarding the MTs arrangement in hydrotropism.
- More information regarding the isolated genes, CLO4 and FAB2 is required in order to possibly relate them with root hydrotropism as an ABA-regulated process. Moreover, the two candidate proteins have an opposing effect in ABA signaling pathways that needs to be rationalized.
Author Response
Comments –
- ktn1 mutants are shown to display significant reduction of hydrotropic root bending upon hydrostimulation, rather normal gravitropic growth but overall have significant reduced primary root growth. Lines 180-182 – “The root growth of ktn1-4 and ktn1-6 seedlings is also much slower than that of WT (Fig. 1A and B). These results suggested that microtubule dynamics is critical for hydrotropic responses but not for gravitropic responses in Arabidopsis roots”. The reduced root growth upon control conditions suggests that the ktn1 mutation have more global effect on root growth that logically also affects hydrotropism. The normal gravitropism might indicate that at the time frame of few hours the attenuated root growth not necessarily affects the early root tropic response. However, it is still not clear whether the reduced hydrotropic response is due to the observed reduced root growth of ktn1 under control conditions, and is not directly related to hydrotropism.
Response: Great thanks for your comments! We have clearly showed that the oryzalin-treated roots showed significantly reduced hydrotropic bending at 11 hours compared with the roots under mock treatment condition (Fig. 1A). However, there were no significant differences in root growth between the 100 nM oryzalin treated and untreated roots within that 11 hours, suggesting that the attenuated hydrotropic bending was not caused by retarded root growth. For ktn mutants that reduced root growth, ktn1-4 and ktn1-6 seedlings displayed significantly reduced root hydrotropic response compared with that of WT (Fig. 2A), but the root gravitropic curvatures in ktn1-4 and ktn1-6 seedlings were not significantly influenced (Fig. 2B). Therefore, our conclusion is that microtubule dynamics is critical for hydrotropic responses but not critical for gravitropic responses, at least at 11 hours, in Arabidopsis roots.
- Treatment with oryzalin (0, 100 nM) of WT seedlings showed no significant effect over root hydrotropism during the initial 4 hrs of stimulation. This was shown by treating the seedlings while exposing the roots to the chemical and moisture gradient simultaneously. It would be more informative if the seedlings were pre-treated with oryzalin prior to hydrostimulation such that the effect on MTs arrangement will be from hydrostimulation start and might provide more insights regarding the MTs arrangement in hydrotropism.
Response: The root is gradual response to the hydrostimulation in a time-course of 24 hours. Therefore, we compared root hydrotropism exposing to the chemical and moisture gradient simultaneously with that of exposing to the moisture gradient alone, which will provide a meaningful data for the research purpose.
- More information regarding the isolated genes, CLO4 and FAB2 is required in order to possibly relate them with root hydrotropism as an ABA-regulated process. Moreover, the two candidate proteins have an opposing effect in ABA signaling pathways that needs to be rationalized.
Response: We have provided more information on these two genes CLO4 and FAB2 at the revised manuscript between lines 261 and 267. “On the one hand, the mRNA level of CLO4 (Caleosin 4) is down-regulated by ABA and salt stress (38). clo4 mutant were more tolerant to drought than the wild type, whereas CLO4-overexpressing (Ox) lines showed hyposensitive to those stresses (38). On the another hand, the mRNA level of FBA2 (Fructose 1,6-biphosphate aldolase 2) is up-regulated following exposure to exogenous ABA, high salinity and drought, and may play key roles in ABA and stress signaling in plants (39). Meanwhile, the Tyr phosphorylation level of FBA is modulated by ABA (40).”
Round 2
Reviewer 2 Report
I thank the authors for addressing all comments and find the latest version as suitable for publication
Cheers